# Diet and Respiratory Infections: Specific or Generalized Associations?

**DOI:** 10.3390/nu14061195

**Published:** 2022-03-11

**Authors:** Thanh-Huyen T. Vu, Linda Van Horn, Chad J. Achenbach, Kelsey J. Rydland, Marilyn C. Cornelis

**Affiliations:** 1Department of Preventive Medicine, Northwestern University Feinberg School of Medicine, 680 N Lake Shore Drive, Suite 1400, Chicago, IL 60611, USA; huyenvu@northwestern.edu (T.-H.T.V.); lvanhorn@northwestern.edu (L.V.H.); c-achenbach@northwestern.edu (C.J.A.); 2Department of Medicine, Northwestern University Feinberg School of Medicine, Chicago, IL 60611, USA; 3Research and Information Services, Northwestern University, Evanston, IL 60208, USA; kelsey.rydland@northwestern.edu

**Keywords:** nutrition, coffee, tea, dietary behaviors, epidemiology COVID-19, respiratory infections, pneumonia, influenza

## Abstract

**Background**: Based on our recently reported associations between specific dietary behaviors and the risk of COVID-19 infection in the UK Biobank (UKB) cohort, we further investigate whether these associations are specific to COVID-19 or extend to other respiratory infections. **Methods**: Pneumonia and influenza diagnoses were retrieved from hospital and death record data linked to the UKB. Baseline, self-reported (2006–2010) dietary behaviors included being breastfed as a baby and intakes of coffee, tea, oily fish, processed meat, red meat (unprocessed), fruit, and vegetables. Logistic regression estimated the odds of pneumonia/influenza from baseline to 31 December 2019 with each dietary component, adjusting for baseline socio-demographic factors, medical history, and other lifestyle behaviors. We considered effect modification by sex and genetic factors related to pneumonia, COVID-19, and caffeine metabolism. **Results**: Of 470,853 UKB participants, 4.0% had pneumonia and 0.2% had influenza during follow up. Increased consumption of coffee, tea, oily fish, and fruit at baseline were significantly and independently associated with a lower risk of future pneumonia events. Increased consumption of red meat was associated with a significantly higher risk. After multivariable adjustment, the odds of pneumonia (*p* ≤ 0.001 for all) were lower by 6–9% when consuming 1–3 cups of coffee/day (vs. <1 cup/day), 8–11% when consuming 1+ cups of tea/day (vs. <1 cup/day), 10–12% when consuming oily fish in higher quartiles (vs. the lowest quartile—Q1), and 9–14% when consuming fruit in higher quartiles (vs. Q1); it was 9% higher when consuming red meat in the fourth quartile (vs. Q1). Similar patterns of associations were observed for influenza but only associations with tea and oily fish met statistical significance. The association between fruit and pneumonia risk was stronger in women than in men (*p* = 0.001 for interaction). **Conclusions**: In the UKB, consumption of coffee, tea, oily fish, and fruit were favorably associated with incident pneumonia/influenza and red meat was adversely associated. Findings for coffee parallel those we reported previously for COVID-19 infection, while other findings are specific to these more common respiratory infections.

## 1. Introduction

Lower respiratory tract infections are among the leading causes of death and illness in people of all ages globally [1]. According to the 2015 Global Burden of Disease study, 55% of ~2.7 million lower respiratory tract infection–related deaths were attributable to pneumococcal pneumonia, and of ~100 million lower respiratory tract infection illnesses, 10.4% were attributable to influenza [1]. Influenza was also associated with more than 14% of global acute respiratory hospital admissions among adults [2]. Knowledge of potentially modifiable risk factors for these respiratory infections would, thus, have important public health implications as it may inform strategies for reducing the burden of these conditions.

The immune system plays a crucial role in the susceptibility and response to infectious diseases [3]. Diet and nutrition are modifiable factors implicated in immunity [4,5,6] and infectious disease acquisition and severity [3,7]. Early-life nutrition, breastfeeding in particular, has been associated with lower rates of asthma, influenza, and other respiratory infections through its impact on the immune system in both infancy and adulthood [8]. We recently reported that being breastfed as a baby, as well as dietary intakes of more coffee and vegetables but less processed meat, was independently associated with lower odds of COVID-19 infection [9]. Whether these associations are specific to COVID-19 or general to viral or bacterial respiratory infections is unclear. Moreover, genetics also plays a role in respiratory infection susceptibility and diet response [10]. For example, genome-wide association studies have identified specific loci associated with pneumonia and COVID-19 susceptibility [11,12]. Whether specific diet behaviors strengthen or weaken these associations (i.e., interaction) is unknown.

The current study expands our previous study of COVID-19 by further investigating whether the same diet behaviors associate with other highly common respiratory infections, including pneumonia and influenza, that can have a fatal impact within certain subgroups. We additionally examine whether genetic susceptibility modifies these associations for the first time.

## 2. Materials and Methods

### 2.1. UK Biobank (UKB)

The UKB includes data from over 500,000 participants aged 37–73 years at 22 centers across England, Wales, and Scotland. Details of the study methods and data collection have been described previously [13] and are available on the Showcase tab of the UKB website [14]. Briefly, in 2006–2010 (baseline), participants were physically assessed and measured for health and risk factors, as well as blood sampling, and agreed to follow up on their health status. UKB ethical approval was from the National Research Ethics Service Committee North West–Haydock (approval letter dated 17 June 2011, Ref. 11/NW/0382). All study procedures were performed in accordance with the World Medical Association Declaration of Helsinki ethical principles for medical research. The current analysis was approved under the UKB application #21394 (PI, M.C.C).

### 2.2. Pneumonia and Influenza Diagnoses

Pneumonia and influenza diagnoses and date of diagnoses were retrieved from hospital admission and death record data linked to the UKB. Diagnoses were based on the International Classification of Diseases (ICD) version 10 and/or 9: J09–J11, 487 for influenza (including influenza due to identified influenza virus, with or without pneumonia and/or other manifestations, and influenza with unidentified virus), and J12–J18, 480-486 for pneumonia (including bacterial or other viral pneumonia that were not elsewhere classified) [15]. Our primary event of interest was any diagnosis (i.e., primary or secondary cause) of pneumonia or influenza occurring after the baseline examination until 31 December 2019 (to minimize the effects of COVID-19 and new public health measures put into place).

### 2.3. COVID-19 Diagnosis and Analysis Sample (Vu et al. 2021)

The criteria for the diagnosis of COVID-19 in the UKB based on tests collected between 16 March and 30 November 2020, and our analytical sample (*n* = 37,988) have been described in detail in our previous report [9] and are briefly summarized in the Appendix A.

### 2.4. Baseline Dietary Data

Participants self-reported their usual intake of 17 pre-selected foods and beverages at baseline using a touchscreen food frequency questionnaire (FFQ) [16]. The details of the questions and possible answers for these dietary behaviors were previously reported [9]. Briefly, participants were asked to report their food consumption by number of pieces/tablespoons/cups of each item or to choose one of several pre-specified frequency categories. Vegetables, fruit, oily fish, processed meat, red meat, tea, and coffee have been implicated in immunity and were targeted for association with COVID-19 in our previous report [9] and, therefore, further considered in the current study. Participants were also asked to respond yes, no, or do not know to whether they were breastfed as a baby.

Consumption of coffee (any type) and tea (black, green) was categorized as none or <1 (referent), 1, 2–3, and ≥4 cups/day. Processed meat, red meat (unprocessed beef, lamb/mutton, or pork), fruit intake (fresh, dried), and vegetable intake (cooked, raw) were categorized as lowest quartile/Q1 (referent), Q2, Q3, and Q4 (servings/day). Oily fish (e.g., sardines, salmon, mackerel, or herring) consumption was initially categorized in quartiles of servings/day; then, Q3 and Q4 were combined because the cut-points to define Q3 and Q4 were similar.

### 2.5. Genetic Data and Single-Nucleotide-Polymorphism (SNP) Selection

All UKB participants were genotyped using genome-wide arrays. Quality control and SNP imputation were performed centrally by the Wellcome Trust Centre for Human Genetics as described elsewhere [17]. We excluded sample outliers based on heterozygosity and SNP missingness, participants with sex discrepancies between the self-reported and X-chromosome heterozygosity, and those potentially related to other participants, based on estimated kinship coefficients for all pairs of samples. We limited the genetic analysis to individuals of British–European (EUR, ~96% of sample) genetically inferred ancestry based on a recent principal component analysis by Pan-UKB Consortium [18,19]. Efforts to investigate other genetic ancestries were limited by sample size and number of incident cases of influenza/pneumonia.

For COVID-19, we selected nine common (minor allele frequency >0.05 in EUR) GWAS SNPs for COVID susceptibility and severity [11,20,21,22] (Appendix A). To complement the *ABO* susceptibility locus, we additionally inferred blood type (O, A, B, AB) using three SNPs (rs8176747, rs41302905, and rs8176719) as described previously [21,22,23]. For pneumonia susceptibility, we selected common GWAS SNPs near *SUCNR1* (rs11708673) [24] and the *HLA* class I region (rs3131623) [25]. For dietary caffeine (tea and coffee), we selected two SNPs with the largest effect sizes in GWAS of caffeine metabolites: rs2472297 (near *CYP1A2*) and rs6968554 (near *AHR*) [26]. Robust SNPs for influenza and other dietary behaviors of interest have not been identified.

### 2.6. Other Covariates

Covariates for the current analysis were measured at baseline. Age, sex, race/ethnicity (assessed as White/Asian/Black/Mixed—Others), education (six qualification classes), income (four levels), employment status (employed/retired/other), physical activity (quartiles of moderate or vigorous activities, min/day), type of accommodation lived in (house/apartment/other) and number of co-habitants (1, 2, 3, or ≥4), smoking behaviors (never/past/current), and current health status (excellent/good/fair/poor) were self-reported using the touchscreen. The Townsend Deprivation Index, with higher scores representing higher deprivation, was derived from participants’ census data and postal codes and assessed as quartiles. Body mass index (BMI) was calculated (as weight/height in meters squared) using height and weight measured at the assessment center and categorized as BMI < 25, 25 ≤ 30, and ≥30 kg/m^2^. Participants self-reported their history of diabetes (yes/no), heart disease (yes/no), and hypercholesterolemia or hypertensive medication use (yes/no) using the touchscreen. A history of pneumonia/influenza (yes/no) was derived from self-report at baseline or hospital records (diagnosis date before baseline visit).

### 2.7. Analysis Samples

Of 502,633 UKB participants, 31,780 participants with missing baseline data on dietary behaviors and covariates were excluded, leaving 470,853 participants for the main analysis (Appendix A). For sensitivity analyses, we used the same analysis sample that was previously used for the COVID-19 outcome, hereafter referred to as the COVID-19 analysis sample (*n* = 37,988) [9]. Effect modification by blood type and genotypes was restricted to genetically inferred EUR participants. Thus, the genetic subsample included 335,205 participants for the main analysis sample (pneumonia/influenza) and 26,919 participants for the COVID-19 analysis sample.

### 2.8. Statistical Analysis

All analyses were performed using SAS (SAS Institute Inc., Cary, NC, USA). We used logistic regression models to examine the associations between each diet behavior and any pneumonia or influenza occurring after the baseline examination until 31 December 2019 (Model 1). We adjusted for all covariates mentioned above, including demographic (age, sex, race, number of co-habitants) and socio-economic status (Townsend Deprivation Index, education, employment status, income, type of accommodation live in), health behaviors (physical activity, smoking, BMI levels), and medical conditions (self-rated health; hypercholesterolemia/hypertensive medication use; and history of diabetes, heart disease, and pneumonia/influenza). Model 2 was similar to Model 1 but with dietary behaviors assessed mutually. Secondary Cox proportional hazards models were also employed; participants were considered at risk for infection from baseline (2006–2010) and were followed up until the date of first diagnosis, death, loss to follow up, or 31 December 2019, whichever came first. Statistical significance was defined as *p* < 0.05. No adjustments were made for multiple testing as all tests were a priori.

To address concerns of selection bias that may limit comparisons between pneumonia/influenza results and our previous COVID-19 results, we conducted sensitivity analyses using the COVID-19 analysis sample (*n* = 37,988). Specifically, we first repeated logistic regressions described above for the main analysis. Secondly, we further restricted analyses to events occurring between 1 March and 31 December 2019. We limited these sensitivity analyses to pneumonia since there were too few cases of influenza in this subsample. Moreover, because pneumonia might be a complication of other diseases, we repeated our primary analysis considering only pneumonia cases reported as a primary cause of hospitalization or death.

For significant pneumonia/influenza–diet associations, we screened for effect modification (interaction) by sex, blood type, and each pneumonia SNP by including the cross-product term of each dietary behavior (e.g., tea consumption, cups/day) and the interacting variable in multivariable regression models. For the significant diet–COVID19 associations we previously reported [11], we now also screened for effect modification by blood type and COVID-19 SNPs. Independent of outcome and exposure, we tested a total of 14 potential effect modifiers, which we used to derive a global statistical significance threshold for interactions of *p* < 0.003 (α/14 tests).

## 3. Results

### 3.1. Participant Characteristics

Characteristics of the main analysis sample, stratified by sex, are presented in Table 1. Compared to females, males were more likely to be employed and current smokers; to have higher education, income, BMI, and comorbidities; and to consume more processed meat and red meat and less fruit and vegetables.

### 3.2. Dietary Behaviors and Risk of Pneumonia/Influenza

After up to 11 years of follow-up, 18,738 participants (3.98%) had at least one pneumonia diagnosis during the follow-up period, and 1120 participants (0.24%) had at least one influenza diagnosis. After adjusting for socio-demographic and medical and lifestyle factors (Model 1), consumption of coffee, tea, oily fish, and fruit was significantly associated with lower odds of having pneumonia, while consumption of processed meat and red meat were associated with higher odds of pneumonia (Table 2). With all dietary factors assessed mutually (Model 2), the pneumonia–processed meat association was no longer significant; other associations were only slightly attenuated but remained statistically significant (*p* ≤ 0.001). The odds of pneumonia were lower by 6–9% when consuming 1–3 cups of coffee/day (vs. <1 cup/day), 8–11% when consuming 1+ cups of tea/day (vs. <1 cup/day); 10–12% when consuming oily fish in higher quartiles (vs. lowest quartile—Q1); and 9–14% when consuming fruit in higher quartiles (vs. Q1). In contrast, the odds of pneumonia were 9% higher for individuals in the fourth quartile of red meat intake (vs. Q1). The latter association was not attenuated in a post hoc analysis with further adjustment for iron-supplement use (yes/no).

Similar results were observed when pneumonia was defined as the primary cause of hospitalization or death (10,343 cases) (data not shown). Results from Cox proportional hazards models are presented in Appendix A and are consistent with findings from logistic regression models described above. Patterns of associations with the risk of influenza (Appendix A) were similar to those reported for pneumonia. However, only associations for tea and oily fish achieved statistical significance.

Table 2 also presents the corresponding results for COVID-19 infection as previously reported [9]. To better align our current results for pneumonia/influenza to these previous results (see Materials and Methods, Section 2), we performed a sensitivity analysis using the same COVID-19 sample. In general, participants in the main analysis sample (*n* = 470,853) tended to have higher measures of socio-economic status, consumed more coffee or tea, and reported better health and fewer comorbidities than those in the COVID-19 analysis sample (*n* = 37,988) (Appendix A). Patterns of associations with pneumonia were similar but attenuated when analyses were restricted to the COVID-19 analysis sample, which included 2187 cases of pneumonia (5.76%). Similar patterns of associations were also observed with a further restriction on date (occurring between 1 March and 31 December 2019) of pneumonia diagnoses (459 cases (1.21%)). However, only results for oily fish remained statistically significant (data not shown).

### 3.3. Effect Modification

We observed a significant interaction (*p* = 0.001) between fruit intake and sex for risk of pneumonia. In general, patterns of associations were similar for men and women, but the favorable association between fruit intake and the risk of pneumonia was stronger in women than in men (Appendix A). Most pneumonia- and COVID-19-related SNPs were significantly associated with their corresponding infection traits (Appendix A). Individuals with blood type A or AB had higher odds of COVID-19 than those with blood type O (Appendix A). No genetic factor modified the diet–pneumonia associations described above or the diet–COVID associations we previously reported (*p* > 0.003 for interactions).

## 4. Discussion

In the current study, consumption of coffee, tea, fish, and fruit was independently associated with a lower risk of future pneumonia/influenza events. Consumption of (unprocessed) red meat was associated with a higher risk. These associations were not modified by genetic susceptibility. Our new findings for coffee paralleled those we reported previously for COVID-19 infection in the same cohort [9]. New findings for tea, fruit, and red meat appeared to be specific to pneumonia/influenza, while previous findings for vegetables, processed meat, and breastfeeding were specific to COVID-19.

Pneumonia is a lung infection caused by bacteria, viruses, or fungi and is an indicator of infection severity. In the UK, pneumonia is the leading lung disease requiring hospital admission, with nearly 30 thousand deaths each year [27]. Age, smoking, environmental exposures, malnutrition, previous or existing respiratory conditions, functional impairment, and immunosuppressive therapy are important risk factors for community-acquired pneumonia (CAP) in adults [28]. Healthcare-associated pneumonia (HCAP) is distinct from CAP and has a higher case fatality rate [29]. Influenza is a virus that typically circulates in a seasonal pattern; up to 20% of the population is infected in any given year [30]. Most individuals experience moderate and short-term respiratory symptoms, while others might experience severe respiratory distress or other complications such as pneumonia. Children, elderly, pregnant mothers, and individuals with metabolic, neurological, and immune-suppressing conditions are at higher risk of severe outcomes [31,32]. Some of these risk factors for pneumonia and influenza overlap with those of COVID-19 [33,34,35]; others are unique. For example, children are less likely to present with severe COVID-19 symptoms, but they are at high risk for influenza and pneumonia [36,37]. Distinguishing risk factors that generalize to all respiratory infections in terms of acquisition (any disease) and severity (pneumonia) from specific risk factors may provide insight to the pathophysiology of these conditions and better inform public health guidelines. In this spirit, the current study investigated dietary behaviors; comparing associations with pneumonia (infection severity) and influenza (infection acquisition) to those we previously reported with COVID-19 [9]. To our knowledge, this is also the first study to examine the role that genetics may play in modifying the relationship between these respiratory infections and dietary behaviors. When interpreting our results, it is important to consider that by using hospital and death records our cases of pneumonia/influenza will generally be at the more severe end of the clinical spectrum. Pneumonia is a complication of severe influenza, COVID-19, and other respiratory conditions [31,32,38,39]; we did not separate HCAP and CAP in the current study. Primary care data were only available for about 40% of the UKB, and we chose not to use these data to avoid case-ascertainment bias. Moreover, such data would have likely captured mild cases.

Coffee and tea are important sources of dietary caffeine, a drug more commonly known for its psychostimulant effects. However, caffeine and its methylxanthine metabolites present with other potentially immunomodulatory properties [40,41]. Experimental studies of caffeine favor inhibitory effects on the proliferation of stimulated lymphocytes, activity of macrophages and natural killer cells, and levels of anti-inflammatory cytokines [40]. Caffeine and its metabolite theophylline are also bronchodilators and may impact respiratory infections indirectly [42]. Many of these effects are likely mediated by caffeine’s ability to antagonize adenosine receptors [43]. Caffeine is also a ligand of several taste 2 receptor (TAS2R) subtypes, which mediate bitter taste perception but also have other newly realized roles in extra-oral physiological processes impacting respiratory health and both innate and adaptive immunity [44,45]. Coffee and tea also have unique profiles of other constituents that may impact immunity differently [46,47,48]. Catechins, theaflavins, and theanine are examples of constituents unique to tea; chlorogenic acid, diterpenes, and Maillard reaction products are unique to brewed coffee [49,50]. We previously reported a lower risk of COVID-19 infection with coffee consumption in the UKB [9]. In the current analysis of pneumonia/influenza, coffee consumption was associated with a lower, albeit modest, risk and only with 1 to 3 cups/day. This non-linear association may suggest a protection threshold but may also be a result of confounding factors correlated with very high coffee intake. Regardless, the findings suggest the mechanism (whether causal or confounded) linking coffee consumption to lower pneumonia/influenza risk is different than that linking coffee to COVID-19 risk; an argument strengthened by the use of the same cohort and similar statistical models. In the UKB, habitual consumption of one or more cups of tea per day was associated with a ~10% lower risk of pneumonia/influenza compared to <1 cup/day. Because we previously observed a similar but non-significant trend for COVID-19 risk in the UKB [9] we cannot, with certainty, rule out a potential generalized benefit of tea drinking on respiratory infections. Human clinical and observational data support the benefits of tea and tea catechin consumption against influenza infection and acute upper respiratory tract infections [51]. Green tea is especially high in catechins, and experimental evidence supports a protective effect of both against influenza; inhibition of viral hemagglutinin- and neuraminidase-mediated functions and replication inhibition are proposed mechanisms [52]. In a subsample of UKB participants providing more detailed tea data, we previously reported that black tea (mean intake ~2 cups/day) was much more commonly consumed than green tea (mean intake <1 cup/day) [53], and thus, other constituents of tea may also underlie the benefits of tea drinking against respiratory infections. Most epidemiological studies of coffee/tea consumption and respiratory-disease-related mortality report beneficial or neutral associations [54,55,56,57,58,59,60,61,62].

In the current study, consumption of at least 0.07 servings/day of oily fish (e.g., sardines, salmon, mackerel, herring) was associated with ~10% lower risk of pneumonia/influenza. The same dietary factor was not associated with COVID-19 infection in the UKB [9]. Oily fish is a unique source of omega 3 fatty acids including EPA and DHA. Several lipid immune mediators are synthesized by these and other long-chain polyunsaturated fatty acids [63]. EPA and arachidonic acid compete for the lipoxygenase and cyclooxygenase pathways for the synthesis of eicosanoids, lipid mediators typically involved in inflammation activation [64]. EPA generates eicosanoids that are less proinflammatory than those generated by arachidonic acid and also suppresses the production of the proinflammatory cytokines interleukin-1β and tumor necrosis factor-α [63].Omega-3s are also precursors of resolvins, protectins, and maresins, which are contra-regulators of proinflammatory mediators critical for resolving inflammation [63,65,66]. Julkunen et al. [67] recently reported a positive correlation between non-fasting plasma levels of omega-3 fatty acids and both COVID-19 and severe pneumonia in a subset of UKB participants; thus a benefit of fish-derived fatty acids on respiratory infections more generally cannot be ruled out. Previous epidemiological studies of dietary omega-3 fatty acid and CAP risk are conflicting [68,69,70]. There have been recent calls for clinical trials of intravenous high-dose fish oil lipid emulsions in hospitalized COVID patients [71]. Our findings and those of others would suggest more human evidence is warranted before initiating such trials.

Fruits and vegetables are rich dietary sources of vitamins, folate, fiber, and phytochemicals—constituents with anti-inflammatory, antibacterial, and antiviral properties [72,73,74,75,76]. In the current study, higher consumption of fruit (fresh and dried), but not vegetables, was associated with a lower risk of pneumonia/influenza. These findings are opposite to those we reported for COVID-19: higher vegetable, but not fruit, consumption was associated with a lower risk [9]. While fruits and vegetables share several health benefits, the specific bioactive compounds in fruits and in vegetables can vary [77] and our findings suggest they impact respiratory-infection-specific pathways. Narrowing in on the specific bioactive substances in fruits that underlie their protective effect against pneumonia/influenza was, therefore, challenging in the current study. Future studies that integrate specific biomarkers of fruit intake would be more informative on these mechanisms.

A different pattern of risk for pneumonia/influenza and COVID-19 also emerged from analyses of processed meat and (unprocessed) red meat. Higher consumption of red meat (at least 0.35 servings/day), but not processed meat, was associated with an increased risk of pneumonia. Higher consumption of processed meat, but not red meat, was associated with an increased risk of COVID-19 [9]. These findings suggest that non-meat factors of red meat and processed-meats associate differently with pneumonia/influenza and COVID-19. Recently, Papier et al. [78] performed a comprehensive analysis of meat consumption and 25 common diseases in the UKB. Increased unprocessed red meat and processed meat intake were each associated with a higher risk of pneumonia (J18). Our study included an expanded definition of pneumonia (J12–J18), which might explain these different results. Papier et al. [78] also reported that increased intake of red meat, but not processed meat, was associated with a lower risk of iron deficiency anemia. Iron deficiency is caused by inadequate nutritional iron intake, impaired iron absorption, increased iron utilization, and blood losses. Effects of iron status on infection susceptibility are not clear and likely vary according to age, setting, and type of infection [79,80]. Indeed both iron deficiency and iron supplementation may increase infection risk or exacerbate existing infections [79,80]. In our post hoc analysis, iron-supplement use did not attenuate the association of red meat intake and pneumonia. Further independent studies are still needed to better determine whether red meat correlates with pneumonia/influenza through iron pathways. The contents of saturated fat, salt, perseverative, and additives are higher in processed meat relative to those in unprocessed red meat, which may underlie the specific relationship between processed meat and COVID-19 [9].

Humans differ in their susceptibility to infectious disease due, in part, to variation in the immune response [81]. Immune pathway genes activated in response to influenza partly overlap those triggered by other single-stranded RNA viruses including COVID-19 [82,83,84]. These immune responses also vary by ancestry and stress the key role played by genetics in shaping population differences in immune responses [81,83,85]. In the current study, we confirmed many of the previously reported COVID–SNP and pneumonia–SNP associations [11,12,20,21,22,24,25], but none of these variants modified associations between COVID/pneumonia and dietary behaviors. Moreover, genetic variation in caffeine metabolism did not modify associations with coffee or tea, suggesting caffeine is unlikely mediating these associations. Although our sample size was large, we cannot rule out interactions with small effect sizes. We acknowledge that the presence of interactions, small or large, may further inform mechanisms but those of small effect sizes will have limited clinical significance.

Strengths of this study include the large sample size, detailed health, lifestyle, and nutrition data, and ongoing follow-up. The ability to compare identical risk factors across conditions with similar statistical models in the same cohort is also a strength. However, in addition to the limitations discussed above, other study limitations warrant mentioning. First, the pneumonia/influenza and COVID-19 samples for analysis present with different participant characteristics, and therefore, differential associations may be a result of selection bias. The UKB cohort is also not representative of the sampling population, with evidence of a ‘healthy volunteer’ selection bias [86]. Second, diet assessment tools are generally prone to measurement error, and therefore, effect sizes may be imprecise [13,87]. Third, the definition of pneumonia and influenza based only on ICD 10/9 codes may lead to an imprecise number of actual cases. However, administrative data have been widely used as a valid method for ascertaining pneumonia cases retrospectively, especially in the hospital setting [88]. Moreover, any possible misclassification on identifying cases using administrative data would be similar across exposure strata (non-differential). Finally, the current study is observational; therefore, we cannot discount the possibility of residual confounding or infer causality.

## 5. Conclusions

In summary, consumption of coffee, tea, oily fish, and fruit were favorably associated with incident pneumonia/influenza and red meat was adversely associated in the UKB. Some of these new findings overlap with those we previously reported for COVID-19 infection and, thus, advocating specific dietary behaviors may impact susceptibility to respiratory infections more generally. This notion, however, warrants more investigation since the spectrum of respiratory infections extends beyond pneumonia/influenza.

## Figures and Tables

**Table 1 nutrients-14-01195-t001:** Baseline characteristics of the UK Biobank.

Baseline Characteristics ^a^	Male	Female
Number of Persons	213,805	257,048
Age, year, mean (sd)	56.78 (8.18)	56.30 (8.00)
Townsend Deprivation Index, mean (sd)	−1.37 (3.09)	−1.40 (3.00)
White/British ^b^	202,928 (94.91)	243,975 (94.91)
Household income, GBP < 18,000	38,076 (17.81)	51,710 (20.12)
College or university degree	73,887 (34.56)	81,234 (31.60)
Currently employed	130,234 (60.91)	142,124 (55.29)
Lived in a house	191,323 (89.48)	232,716 (90.53)
Number of co-habitants ≥ 4	43,910 (20.54)	45,113 (17.55)
Current smoker	26,379 (12.34)	23,298 (9.06)
BMI (kg/m^2^), mean (sd)	27.81 (4.22)	27.05 (5.17)
Physical activity, minutes/day, mean (sd)	83.14 (108.22)	68.85 (83.95)
Poor overall health rating	10,609 (4.96)	9687 (3.77)
Cholesterol medication use	48,686 (22.77)	32,033 (12.46)
Blood pressure medication use	52,080 (24.36)	44,441 (17.29)
History of diabetes	14,635 (6.85)	9495 (3.69)
History of heart disease	18,028 (8.43)	8535 (3.32)
History of pneumonia	4441 (2.08)	4270 (1.66)
History of influenza	44 (0.02)	56 (0.02)
Breastfed as baby	115,990 (54.25)	146,232 (56.89)
Coffee, cups/day, mean (sd)	3.39 (1.57)	3.18 (1.53)
Tea, cups/day, mean (sd)	4.14 (1.69)	4.10 (1.70)
Oily fish, servings/day, mean (sd)	0.16 (0.15)	0.16 (0.15)
Processed meat, servings/day, mean (sd)	0.27 (0.22)	0.16 (0.17)
Red meat, servings/day, mean (sd)	0.32 (0.22)	0.28 (0.20)
Fruit (fresh/dried), servings/day, mean (sd)	2.74 (2.58)	3.32 (2.56)
Vegetables (cooked/raw), servings/day, mean (sd)	0.78 (0.58)	0.85 (0.54)

Abbreviation: BMI, body mass index; sd, standard deviation. ^a^ Data drawn from baseline (2006–2010). Values are numbers (%) unless stated otherwise. ^b^ All differences between male and female participants are significant (*p* < 0.001) except White/British.

**Table 2 nutrients-14-01195-t002:** Dietary behaviors and risk of pneumonia.

Dietary Behavior	Current Analysis				Vu et al. [11]	
	Pneumonia ^b^ (*n* = 470,853)				COVID-19 Infection ^c^ (*n* = 37,988)	
	Model 1 ^a^		Model 2		Model 2	
	OR (95% CI)	*p*	OR (95% CI)	*p*	OR (95% CI)	*p*
Coffee, cups/day						
None or <1 cup	Reference		Reference		Reference	
1 cup	0.90 (0.86, 0.94)	<0.0001	0.91 (0.87, 0.96)	<0.0001	0.90 (0.83, 0.98)	0.015
2–3 cups	0.93 (0.89, 0.97)	<0.0001	0.94 (0.90, 0.97)	0.001	0.90 (0.83, 0.96)	0.003
≥4 cups	1.03 (0.99, 1.08)	0.165	1.00 (0.96, 1.05)	0.963	0.92 (0.84, 0.99)	0.047
Tea, cups/day						
None or <1 cup	Reference		Reference		Reference	
1 cup	0.87 (0.81, 0.93)	<0.0001	0.89 (0.83, 0.95)	<0.0001	0.93 (0.82, 1.04)	0.204
2–3 cups	0.86 (0.82, 0.90)	<0.0001	0.88 (0.84, 0.92)	<0.0001	0.93 (0.85, 1.01)	0.078
≥4 cups	0.90 (0.87, 0.94)	<0.0001	0.92 (0.88, 0.96)	<0.0001	0.98 (0.90, 1.06)	0.543
Oily fish, servings/day						
Q1 (0–<0.07)	Reference		Reference		Reference	
Q2 (0.07–<0.14)	0.88 (0.83, 0.92)	<0.0001	0.88 (0.84, 0.93)	<0.0001	0.94 (0.86, 1.03)	0.183
Q3 and 4 (≥0.14)	0.87 (0.83, 0.92)	<0.0001	0.90 (0.85, 0.94)	<0.0001	0.98 (0.90, 1.07)	0.654
Processed meat, servings/day						
Q1 (0–<0.07)	Reference		Reference		Reference	
Q2 (0.07–<0.14)	0.97 (0.91, 1.03)	0.258	0.97 (0.91, 1.04)	0.355	1.05 (0.93, 1.19)	0.410
Q3 (0.14-–<0.43)	1.02 (0.96, 1.09)	0.457	1.02 (0.95, 1.09)	0.627	1.09 (0.97, 1.24)	0.155
Q4 (≥0.43)	1.07 (1.00, 1.14)	0.038	1.05 (0.98, 1.12)	0.188	1.14 (1.01, 1.29)	0.036
Red meat, servings/day						
Q1 (0–<0.21)	Reference		Reference		Reference	
Q2 (0.21–<0.28)	0.99 (0.94, 1.04)	0.599	1.01 (0.96, 1.06)	0.855	0.95 (0.87, 1.04)	0.236
Q3 (0.28–<0.35)	1.05 (0.99, 1.10)	0.089	1.06 (1.00, 1.12)	0.048	1.00 (0.90, 1.10)	0.948
Q4 (≥0.35)	1.08 (1.03, 1.13)	0.001	1.09 (1.03, 1.14)	0.001	0.98 (0.89, 1.07)	0.600
Fruit (fresh/dried), servings/day						
Q1 (0–<1.00)	Reference		Reference		Reference	
Q2 (1.00–<2.25)	0.90 (0.85, 0.94)	<0.0001	0.91 (0.87, 0.96)	0.001	1.05 (0.95, 1.16)	0.376
Q3 (2.25–<4.00)	0.84 (0.79, 0.89)	<0.0001	0.86 (0.81, 0.91)	<0.0001	1.02 (0.91, 1.14)	0.762
Q4 (≥4.00)	0.83 (0.79, 0.88)	<0.0001	0.86 (0.81, 0.91)	<0.0001	1.03 (0.92, 1.15)	0.660
Vegetables(cooked/raw), servings/day						
Q1 (0–<0.50)	Reference		Reference		Reference	
Q2 (0.50–<0.67)	0.97 (0.93, 1.01)	0.152	1.00 (0.96, 1.05)	0.972	0.93 (0.85, 1.00)	0.060
Q3 (0.67–<1.00)	0.96 (0.91, 1.01)	0.096	1.00 (0.94, 1.05)	0.895	0.88 (0.80, 0.98)	0.015
Q4 (≥1.00)	0.98 (0.94, 1.03)	0.403	1.03 (0.99, 1.08)	0.182	0.92 (0.84, 0.99)	0.046
Breastfed as a baby						
No	Reference		Reference		Reference	
Yes	0.96 (0.92, 1.01)	0.083	0.97 (0.93, 1.01)	0.165	0.91 (0.85, 0.98)	0.013
Do not know	1.00 (0.96, 1.05)	0.982	1.00 (0.96, 1.05)	0.975	0.98 (0.90, 1.07)	0.696

Abbreviations: OR, odds ratio; CI, confidence interval; Q, quartile. ^a^ Model 1: Adjusted for Townsend Deprivation Index, baseline age, sex, race, education, income, employment status, type of accommodation lived in, number of co-habitants, BMI level, smoking status, physical activity, self-rated health, cholesterol-lowering medication use, antihypertension medication use, history of diabetes, history of cardiovascular disease, and history of pneumonia. Individual diet factors assessed in separate models. Model 2: Adjusted for all covariates listed in Model 1, with all diet factors included in the model (i.e., mutual adjustment). ^b^ Any diagnosis in the hospital database or death records from baseline to 31 December 2019. ^c^ Any confirmed COVID-19 infection (defined as having any positive PCR test result for SARS-CoV-2) between 16 March and 30 November 2020 (Vu et al. [9]).

## Data Availability

The data that support the findings of this study are available from the UK Biobank upon approved request.

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
