# Peer review of "Diet and Respiratory Infections: Specific or Generalized Associations?"

_nutrients, 2022, doi:10.3390/nu14061195_

Round 1
Reviewer 1 Report
This is a large cross-sectional study reporting the association between diet and pneumonia and influenza. The study results support previous literature regarding the beneficial association of coffee, oily fish, and fruit consumption with respiratory infections, focusing in pneumonia and influenza. It also assesses genetic data and association with the studied outcomes.
One of main limitations, that should be further stated in the discussion section, is the definition used for Pneumonia and Influenza, reported medical diagnosis.
Is there any data regarding the molecular diagnosis of Influenza or confirmation?
Is it only possible to include Influenza ICD 10 diagnosis without a positive diagnostic test?
Pneumonia was used in any diagnosis, primary or secondary, however pneumonia might be a complication of other disease and perhaps analyzing the patients that were primarily admitted for pneumonia might show different results.
Is it possible to perform a specific sensitivity analysis for those that were admitted for pneumonia as a primary diagnosis, as it was not possible to differentiate those with health care associated pneumonia and community acquired pneumonia?
Discussion can be further improved.
The main aim of the study is focused on pneumonia/influenza infections, nevertheless most of the second paragraph of discussion is comparing and describing the main risk factors for COVID infection, this could be summarized and presented only the correlated data with this research.
There is no focus on the rational of the potential beneficial effects of caffeine and caffeine metabolism on respiratory infections. There have been previous reports from the literature exploring these association in the literature and should be further enlightened in this manuscript.
Similarly, the discussion regarding the potential mechanisms that might mediate the benefit of oily fish and fruit and vegetable with respiratory infection protection is needed
There is also no reference in the discussion regarding the genetic data analyzed, although no significant result was found, it should be discussed balancing potential limitation and strengths of the analysis.
Regarding the possible association with red meat and iron deficiency is there any data regarding anemia or previous use of iron or vitamin supplements? Can this be a confounder?
Is there in the dataset any information regarding use of supplements and probiotics?
In the conclusion and title, it should focus on pneumonia/influenza, as the spectrum of respiratory infection is wider, and the studied data is linked specific with those specific diagnosis. It is difficult to have external validity to all respiratory infections with the obtained study results.
Reviewer 2 Report
The paper by Vu et al. sought to identify whether there was a link between diet and respiratory infections such an influenza and pneumonia. Research also sought to determine whether there was an effect modification by genetics. While the authors are clearly excellent writers and the research question is interesting, I think the manuscript does not adequately address the novelty in the research question.
The way the introduction currently reads is that researchers previously reported that dietary patterns influence the risk of contracting covid-19, and want to see whether these associations exist in flu and pneumonia. That seems like those are just other variables thrown into the model rather than setting up why this research question is just as important in flu and pneumonia.
The last sentence of the introduction also includes that research wanted to determine whether there were genetic influences. There was no explanation throughout the introduction of the gap in the literature with this specific variable, or why it was important to consider in this study. Therefore, I think the introduction needs to be revamped and the novelty needs to be “beefed up”.
While I do not have major qualms about the statistical analyses and the research question is adequately addressed, I ultimately feel the research question and paper written as is, is not novel.
On the CDC website and in a PubMed search, dietary patterns have been investigated in a wide host of immune outcomes, including influenza risk. Perhaps this paper could be written as a short report or rapid communication, and the genetic piece could be properly discussed so that the readership understands that this is novel.
I feel that some of the discussion and the way influenza and pneumonia are discussed (such as lines 239 on) could be incorporated into the discussion so that the reader could understand the disease and the public health impact.
Seems like authors are bringing up their previous reports with covid-19 frequently throughout the paper, which is why it does not seem novel or like a stand-alone paper.
How many of the influenza/pneumonia data overlap with the covid-19 data? I have heard from physicians that cases of covid-19 can be documented as pneumonia and vice versa.
Round 2
Reviewer 2 Report
While the authors took out three sentences in the beginning about covid-19 infections, they did did not add in any additional mechanisms or justification as to why viral or bacterial respiratory infections, specifically pneumonia or influenza, would be impacted by diet.
The introduction still reads like they just wanted to see whether the associations exist with other viruses since associations are present with diet and covid-19 . I still think there should be a few sentences fleshing this out to explain how and why diet may influence susceptibility to influenza or pneumonia. In this reviewer’s opinion, the authors need to rework the introduction so we know why it’s important to look at pneumonia and influenza (even if that means including some of the prevalence data from the discussion to show why the problem is important, and then include why there could be an impact of diet).
The discussion is greatly improved though I still feel it could be condensed.
